# Direct and Indirect Predictors of Burden in Arab-Bedouin and Jewish-Israeli Mothers Caring for a Child with Epilepsy

**DOI:** 10.3390/healthcare11192662

**Published:** 2023-10-01

**Authors:** Idit Joss, Yaacov G. Bachner, Talia Shorer, Zamir Shorer, Norm O’Rourke

**Affiliations:** 1Department of Epidemiology, Biostatistics and Community Health Sciences, Ben-Gurion University of the Negev, Be’er Sheva 8410501, Israel; 2Center for Multidisciplinary Research in Aging, Ben-Gurion University of the Negev, Be’er Sheva 8410501, Israel; 3Pediatric Neurology Unit, Soroka Medical Center, Be’er Sheva 84101, Israel; 4Department of Psychology, Ben-Gurion University of the Negev, Be’er Sheva 8410501, Israel

**Keywords:** caregiver burden, epilepsy, ethnic differences, family care, mothers, path analysis

## Abstract

Objective: Caring for a child with epilepsy poses various psychological, physical and medical challenges; these can lead to caregiver burden. The aim of this study was to identify predictors of burden with mothers caring for a child with epilepsy. Our analyses included sociodemographic (e.g., ethnicity), mental health (e.g., symptoms of anxiety, depression) and physiological factors (e.g., extent of pharmacotherapy). Methods: A total of 168 mothers caring for a child with epilepsy were recruited while attending the Pediatric Neurology Clinic at Soroka Medical Center, Be’er Sheva, Israel. This cross-sectional sample included 130 Jewish-Israeli and 38 Arab-Bedouin mothers who completed parallel questionnaire batteries that included the Zarit Burden Interview and other scales translated and validated in Hebrew and Arabic. We computed path analyses to identify both direct and indirect predictors of caregiver burden. Results: Burden was directly predicted by emotional exhaustion, symptoms of anxiety and (Bedouin) ethnicity. Indirect effects on burden included illness severity (via emotional exhaustion), ethnicity and emotional exhaustion (both via anxiety). That is, both ethnicity and emotional exhaustion directly and indirectly predicted caregiver burden via greater anxiety. Illness severity indirectly predicted symptoms of depression, anxiety and caregiver burden. We found that 55% of epilepsy care burden was predicted by this path model. Conclusions: Bedouin mothers reported greater illness severity, symptoms of depression, anxiety and caregiver burden. Differences between groups in epilepsy severity suggest that less severe cases in the Bedouin community do not come to clinical attention (e.g., are concealed due to stigma). These findings underscore the need for health promotion strategies and interventions for caregivers tailored to account for ethnic and cultural differences.

## 1. Introduction

Epilepsy is a common neurological condition affecting about 50 million people of all ages worldwide. Up to 80% of people with epilepsy live in low- and middle-income countries, which greatly affects their access to education, diagnostic and treatment services [1]. Prevalence also differs, ranging from 0.32 to 0.55% in developed countries and from 0.36 to 4.4% in developing countries [2]. Epilepsy affects 0.5–1.0% of children under 16 years of age [3]. Incidence varies by age, from 0.14% in the first year of life to 0.058% through the following years up to age 10. Cumulative incidence is 0.45% at age 5 and 0.66% at 10 years of age [4].

In addition to seizures, children with epilepsy often present with comorbid conditions. These include educational and cognitive difficulties, medication side effects, social stigma, behavioral disturbances and psychiatric disorders [5,6]. Children with epilepsy are also prone to accidents and injuries [7]. More precisely, children and adolescents with epilepsy incur 18% more bone fractures and 49% more burns than other children [8].

These challenges require parents to oversee many facets of their child’s lives. This includes attending to their medical, psychological and physical health and rehabilitation, all of which must be coordinated with family, social networks and healthcare providers [9]. Epilepsy is a chronic disease that requires long-term treatment which, in turn, affects the social, financial and emotional well-being of families [9]. Accordingly, caregivers experience high stress, unmet personal and health needs, depressive symptoms and, for many, financial burden and psychological burnout [10]. For example, families in the U.S. with a child with epilepsy have additional medical expenses exceeding $9,103.25 USD per year, greater than other disorders with childhood onset such as asthma, diabetes, food allergies and hypertension [11].

Beyond the instrumental demands of epilepsy care and immediate effects, many caregivers receive limited or inconsistent support from social networks and extended families. As a result, many shield their children to limit shame, embarrassment and rejection (e.g., stigma). For many, social interactions are few, which further contribute to limited support, separation and loneliness [12].

Consistent and enduring epilepsy care demands can lead parents to feel overwhelmed. Below, we synthesize the existing literature specific to parents of children with epilepsy and other chronic health conditions beginning in childhood. Some topics such as depression are more widely studied than others (e.g., ethnicity). These variables are included in our path model.

### 1.1. Anxiety and Depression

Care demands create various challenges for parents of children with epilepsy. Caregiver burden is affected by family dynamics and psychological distress [13]. Up to 50% of mothers of children with epilepsy report elevated symptoms of depression and anxiety [14,15]. Mothers caring for sons report greater anxiety than fathers. Married caregivers and parents of children experiencing five or more seizures per year report greater depression compared to unmarried parents and parents of children having fewer than five seizures per year [16].

### 1.2. Emotional Exhaustion

Emotional exhaustion (EE) is defined as feeling overextended and emotionally drained by others [17]. Associated feelings include anger, frustration, guilt, sadness and insecurity, along with physical symptoms including headaches [18]. Parents with chronically ill children experience more EE than parents with healthy children [19,20]. Similar findings have been reported with caregivers of children with chronic neurological illness. With cancer caregivers in Israel, EE predicts depressive symptoms [18]; this has been reported with both secular and orthodox caregivers [21].

### 1.3. Illness Severity and Treatment

About one-quarter of parents report little or no negative impact of epilepsy care. Those caring for children with good seizure control report comparatively few negative effects relative to parents of children with poor seizure control [22]. Poor seizure control impedes children’s ability to learn, grow and develop, similar to children with cerebral palsy and developmental disorders [12]. Symptoms of anxiety and depression increase with the frequency and severity of epilepsy symptoms [23].

### 1.4. Ethnic and Socioeconomic Factors

In addition, and often complicating care of children with epilepsy, are sociodemographic factors such as income and ethnicity. Families with few resources experience greater stress due to medical expenses. Lower socioeconomic status predicts greater maternal anxiety and family stress [24]. Ethnic minorities often live in societies with finite resources and other priorities, which further affects economic means, social support and burden. In Israel, Arab mothers tended to report lower social support and greater symptoms of depression than their Jewish counterparts [25].

Symptoms of epilepsy may also present differently across ethnic groups whose beliefs, values and responses differ, in turn, affecting caregivers’ mental health and well-being [26]. For example, in developing countries like China with limited education and mental health knowledge, it is believed that seizure control and care for comorbidities supersede the well-being of caregivers [27]. This is compounded by misconceptions and mental health stigma [28].

Israel is a diverse and multicultural society. In the south, about a third of the population are Negev Bedouin, who are a minority within the Arab minority (i.e., 15% of 23%) [29]. Today, most Bedouin live in recognized cities and towns; a lower percentage (30%) live in transient desert settlements [30]. Access to healthcare is limited in these unrecognized communities where basic infrastructure such as running water is lacking [31].

### 1.5. Bedouin Society in Israel

The Bedouin Negev are a pastoral, semi-nomadic Muslim society who adhere to traditional, collectivist beliefs and values [31]. For example, both polygamy and consanguineous marriage between first cousins remain common [26], with significant affect maternal health and well-being [32].

Though healthcare is available to all residents of Israel, traditional family structures affect access for the Bedouin Negev [33]. For instance, primary care physicians are often relatives, which can be uncomfortable, especially for women consulting a male cousin; yet seeking opinions outside the Bedouin community is discouraged [34].

### 1.6. Study Objectives

The aim of this cross-sectional study was to identify direct and indirect predictors of epilepsy care burden. Mothers are most often primary caregivers of children with epilepsy [35,36,37] and commonly report high levels of stress, anxiety and depression [38]. We hypothesized that symptom severity, ethnicity, symptoms of depression, anxiety, and emotional exhaustion directly predict burden among Jewish and Bedouin mothers.

## 2. Methods

A total of 168 mothers of children with epilepsy (130 Jewish, 38 Bedouin) were recruited for an extended study of epilepsy care [31,39]. According to our inclusion criteria, we selected women 18+ years of age with a child diagnosed with epilepsy, and able to read and write Hebrew or Arabic.

Prospective participants attended the Pediatric Neurology Clinic, Soroka Medical Center, Be’er Sheva, Israel. Mothers were told that participation was voluntary and that they could terminate at any time. Participants completed the questionnaire battery independently onsite and returned it to a research assistant in a sealed envelope. This study was approved by the Ethics Committee, Soroka Medical Center, Be’er Sheva, Israel (April 2014, SOR-0168-12).

### 2.1. Measures

Caregiver burden was measured using a 12-item version of the Zarit Burden Interview (ZBI) which measures perceived strain due to unpaid family care [40,41]; frequency is reported on a Likert scale ranging from *never* (0) to *nearly always* (4). Though initially developed for dementia research and practice, the ZBI has been widely used across populations and groups of caregivers (0.82 < α < 0.95) [42]. This includes research conducted in Israel using both Hebrew and Arabic versions of the ZBI [43]. Internal consistency of participant responses in this study was high, α = 0.88. Range of scores was 0–44.

Emotional exhaustion was measured using the EE subscale from the Maslach Burnout Inventory [44]. Degree of endorsement to each of five statements is reported using a Likert-type scale ranging from *not at all true* (1) to *very true* (5). Internal consistency of EE scale responses by cancer caregivers in Israel were found to be ideal (α = 0.92) [18]. We obtained the same alpha value (α = 0.92) for EE responses by participants recruited for this study. Range of scores 1–5.

Symptoms of anxiety and depression were measured by the Hospital Anxiety and Depression Scale (HADS) [45]. Originally developed as a screening measure in acute care, the HADS has since been used widely across populations and clinical settings [46]. Responses to 7 depression and 7 anxiety items are reported on a Likert scale ranging from *never* (1) to *nearly always* (4), with good internal consistency reported for both (α = 0.86 & α = 0.89, respectively) [47]. For this study, internal consistency was again high for depression (α = 0.83) and anxiety (α = 0.78). Scores ranged from 7–23 and 7–26, respectively.

We constructed a sociodemographic questionnaire to collect clinical and descriptive information. This enabled us to build an illness severity checklist as an objective index of epilepsy care demands. Items include age of epilepsy onset (0–1, 1–3, 4–8, 9+ years, reversed), epilepsy duration (0–1, 1–3, 4–6, 7+ years), seizure severity (minimal, medium, considerable), seizure duration (seconds, minutes, hours), seizure frequency (yearly, monthly, weekly, daily) and date of last seizure (year, month, week, day before). Higher totals suggest greater epilepsy severity; scores ranged from 7–22.

We also measured the extent of pharmacotherapy based on responses to four questions including number of prescribed seizure medications (1–5), medication side-effects (yes, no), extent of seizure control (full, partial, no) and seizure-related injuries (yes, no). Scores ranged from 2-8 with higher totals indicating more intensive pharmaceutical treatment. 

### 2.2. Statistical Methods

Path analysis was performed for this study as a three-step process [48]. A hypothesized model was first tested in which all independent variables were assumed to be direct predictors of caregiver burden; nonsignificant paths were deleted, and statistically significant paths not initially hypothesized were added if supported by existing research or theory. With six independent variables, a sample of 168 participants was sufficient to identify medium to large effect sizes (where *d* = 0.80, α = 0.05) [49].

Path analysis is an extension of linear regression with three significant advantages. Path analysis allows us to simultaneously predict one or more dependent variables (touched by an arrowhead in path models). Arrows pointing from independent to dependent variables represent significant prediction (i.e., critical ratio values >|1.96|, *p* < 0.05). Path analysis is a multivariate statistical procedure, meaning that all significant paths emerge concurrently (i.e., over and above other statistically significant results).

Path models allow us to identify both direct and indirect predictors of burden [50]. Indirect prediction occurs via other variables (i.e., 2+ pathways between variables). In more complex models, variables can have direct and indirect effects on dependent variables, and indirect effects can be of equal or greater magnitude compared to direct effects (total effects = direct + indirect effects).

Computing path analyses with structural equation modeling (SEM) software allows us to obtain goodness of fit information for the overall model. Good model fit is required to interpret models [49].

In accord with convention, we report three goodness-of-fit indices to assess overall model fit: an incremental index (CFI; Comparative Fit Index); an absolute index (SRMR; Standardized Root Mean Residual); and a parsimonious fit index (RMSEA; Root-Mean-Square Error of Approximation). Ideal SRMR and RMSEA values are less than 0.055 whereas ideal CFI values are greater than 0.95 [48,49]. Descriptive and comparative analyses were performed using SPSS v26, and path analyses were performed using AMOS v26.

## 3. Results

For this study, we recruited 130 Jewish and 38 Bedouin mothers of children diagnosed with epilepsy. On average, these women were 40.36 years of age (SD = 7.90), had completed 13.80 years of education (SD = 2.48) and most rated their economic status as fair. Between group comparisons indicate that Jewish mothers were older, more educated and less religious than their Bedouin counterparts. Groups did not differ in economic status, though Bedouin mothers had more children than their Jewish counterparts. Children with epilepsy were 11.1 years of age on average and had been diagnosed with epilepsy 4.51 years ago (SD = 4.97). Ethnic groups did not differ in age or gender composition. See Table 1.

Significant differences between groups were found for most study measures. Compared to their Jewish counterparts, Bedouin mothers reported greater burden, illness severity, pharmacotherapy and symptoms of depression and anxiety. Bedouin mothers reported greater EE than Jewish mothers, but this difference was not significant between groups. See Table 1.

### Path Analyses

We computed path analyses using the maximum likelihood method of parameter estimation [50]. With six independent variables, our sample of 168 mothers was sufficient to detect medium to large effects, where α = 0.05, and *d* = 0.80 [48].

Goodness of fit was within optimal values for each of the three statistics examined, χ^2^ (*df* = 11) = 12.47, *p* = 0.33. That is, the Comparative Fit Index (CFI = 0.99), the Standardized Root Mean Residual (SRMR = 0.053) and the Root-Mean-Square Error of Approximation (RMSEA = 0.028) were within the ideal limits. The full 90% confidence interval for the RMSEA was within acceptable limits (0 < RMSEA CL_90_ < 0.089). 

Burden was directly predicted by emotional exhaustion, symptoms of anxiety and (Bedouin) ethnicity. Indirect effects on burden included illness severity (via emotional exhaustion), ethnicity and emotional exhaustion (both via anxiety). That is, ethnicity and emotional exhaustion both directly and indirectly predicted care burden via anxiety. Overall, 55% of epilepsy care burden was predicted by this path model, *R*^2^ = 0.55, *p* < 0.01. See Figure 1 and Table 2.

As mentioned, Bedouin mothers reported greater illness severity (*r* = 0.26, *p* < 0.01), symptoms of depression (β = 0.14, *p* < 0.01), anxiety (β = 0.13, *p* < 0.05) and epilepsy care burden (β = 0.25, *p* < 0.01). Illness severity indirectly predicted symptoms of depression, anxiety and burden. Emotional exhaustion indirectly predicted burden (via anxiety) and depressive symptoms (via exhaustion and anxiety).

## 4. Discussion

Caring for a child with epilepsy is often difficult, requiring vigilance and associated stress. This leads many caregivers to feel burdened. Demands of care are even greater for ethnic minority families with few resources and faced with structural barriers. For this study, we identified direct and indirect predictors of burden among Jewish and Bedouin mothers of children with epilepsy.

Jewish mothers recruited for this study were older, more educated, less religious and had fewer children than their Bedouin counterparts. These features reflect deeply-rooted culture differences. Jewish culture is distinct from Arab-Bedouin culture with regard to language, religion, beliefs, customs, lifestyle, family structure, attitudes toward healthcare [50,51,52] and use of medical services [53]. Therefore, it is not surprising that Bedouin mothers reported greater burden and more symptoms of depression and anxiety than their Jewish counterparts. These findings are in accord with Romito and colleagues [54], who concluded that differences in ethnicity, cultural values, beliefs and family systems lead caregivers from different cultures to experience their roles differently.

Emotional exhaustion, symptoms of anxiety, Bedouin ethnicity and illness severity were found to predict caregiver burden directly, indirectly or both directly and indirectly. Of note, depression did not appear to predict burden in this sample. Moreover, mothers reported a relatively low level of depression. This finding is in contrast with previous research reporting both high levels of depression [16] and significant associations between depression and burden among parents (especially, the mothers) of children with epilepsy [39] and those caring for family members with chronic health conditions (i.e., cancer, dementia) [26,55].

Instead, we found that of all study variables, emotional exhaustion was the strongest direct predictor of caregiver burden; moreover, emotional exhaustion indirectly predicted burden via symptoms of anxiety. In fact, the proportion of burden variance explained by emotional exhaustion was 2–3 times greater than that associated with epilepsy severity, (Bedouin) ethnicity and symptoms of anxiety. This suggests that emotional exhaustion is integral to the experience of caregiver burden for mothers of children with epilepsy [56].

This finding is understandable, as mothers of children with epilepsy are always vigilant for harbingers of seizures, dispensing medication and managing cognitive and behavioral difficulties [57]. These demands affect the sleep of epilepsy caregivers as well as their physical and emotional health [56,57,58]. These factors likely contribute to elevated emotional exhaustion.

This appears especially true for Bedouin mothers who reported greater illness severity, symptoms of depression, anxiety and caregiver burden. Differences in epilepsy severity between groups suggest that less severe cases in the Bedouin community do not come to clinical attention. This may be due to misconceptions, limited knowledge and mental health stigma [28], underscoring the added and complex challenges faced by ethnic minority caregivers. In addition to sociodemographic limitations, sociopolitical factors [31] make epilepsy care more challenging for Bedouin mothers.

## 5. Limitations and Future Research

Several limitations of our study should be noted, including its cross-sectional design; no causal associations should be inferred. Second, our sample size was modest, as we recruited participants only from the south of Israel, who did not represent either the Jewish or Arab population. Our findings may not be widely generalizable. Third, Bedouin mothers constituted only a third of our sample. This corresponds to the population of southern Israel, but our findings should be replicated. Future, longitudinal research should be conducted with larger samples other ethnic minorities (e.g., Bedouin in the north of Israel, Arab Christians).

## 6. Conclusions

Despite these limitations, we identified both direct and indirect predictors of epilepsy care burden. Our results suggest that illness severity, symptoms of depression, anxiety and caregiver burden are greater for Bedouin mothers than their Jewish counterparts. These findings underscore the importance of health promotion strategies and interventions for caregivers tailored to account for ethnic and cultural differences [28].

## Figures and Tables

**Figure 1 healthcare-11-02662-f001:**
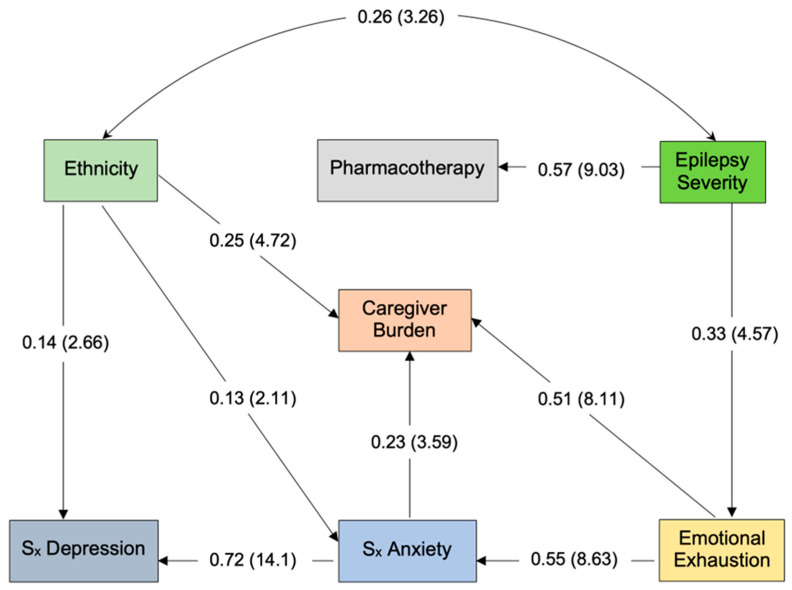
Direct and indirect predictors of epilepsy care burden in Arab-Bedouin and Jewish-Israeli mothers. Note: Parameter estimates expressed as maximum likelihood estimates (standardized solution). Parenthetical numbers indicate significance estimates (CR > 1.96, *p* < 0.05; CR > 2.58, *p* < 0.01).

**Table 1 healthcare-11-02662-t001:** Descriptive and comparative statistics for Jewish Israeli and Arab Bedouin mothers, sociodemographic characteristics and study variables.

		Total (*n* = 168)		Jewish Mothers (*n* = 130)		Bedouin Mothers (*n* = 38)		
Variables	Range	Mean	SD	Mean	SD	Mean	SD	*t* Test
Age of mother	24–57	40.24	7.80	41.72	7.65	34.54	5.44	6.16 **
Years of education	6–24	13.56	2.85	13.99	2.42	12.03	3.50	3.84 ***
Religiosity	1–4	2.29	0.94	2.08	0.93	3.03	0.55	−7.90 ***
Economic status	1–5	2.48	0.82	2.41	0.72	2.70	1.09	1.62
Age of child	1–23	11.14	5.31	11.41	5.29	10.22	5.33	1.21
Number of children	1–11	3.61	1.83	3.32	1.55	4.65	2.32	−3.26 ***
Caregiver burden	0–44	17.98	10.33	16.10	9.57	24.42	10.34	−4.63 ***
Emotional exhaustion	1–5	2.48	1.27	2.41	1.25	2.69	1.36	−1.19
S_x_ of Depression	7–23	13.11	3.80	12.56	3.68	15.00	3.64	−3.60 ***
S_x_ of Anxiety	7–26	16.71	4.42	16.27	4.32	18.21	4.47	−2.42 **
Epilepsy severity	7–22	13.60	3.47	13.11	3.42	15.26	3.15	−3.48 ***
Pharmacotherapy	2–8	4.01	1.68	3.82	1.51	4.68	2.04	−2.44 ***

Note: S_x_ = symptoms, SD = standard deviation. *** *p* < 0.001 ** *p* < 0.01.

**Table 2 healthcare-11-02662-t002:** Predictors of epilepsy care burden in Arab-Bedouin and Jewish-Israeli mothers.

	Epilepsy Severity	Ethnicity	Emotional Exhaustion	S_x_ of Anxiety
**Emotional Exhaustion**				
◦ direct effect	0.33			
◦ indirect effect	0.00			
Total effects	0.33			
**S_x_ of Anxiety**				
◦ direct effect	0.00	0.13	0.55	
◦ indirect effect	0.18	0.00	0.00	
Total effects	0.18	0.13	0.55	
**Pharmacotherapy**				
◦ direct effect	0.25			
◦ indirect effect	0.00			
Total effects	0.25			
**Caregiver Burden**				
◦ direct effect	0.00	0.25	0.51	0.23
◦ indirect effect	0.21	0.03	0.13	0.00
Total effects	0.21	0.28	0.64	0.23
**S_x_ of Depression**				
◦ direct effect	0.00	0.14	0.00	0.72
◦ indirect effect	0.13	0.09	0.40	0.00
Total effects	0.13	0.23	0.40	0.72

## Data Availability

Anonymized data are available from the corresponding author upon request.

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
