# Peer review of "Direct and Indirect Predictors of Burden in Arab-Bedouin and Jewish-Israeli Mothers Caring for a Child with Epilepsy"

_healthcare, 2023, doi:10.3390/healthcare11192662_

Round 1

Reviewer 1 Report

Thanks for letting me review this interesting paper. Here are a few suggestions to improve the manuscript

GENERAL

I suggest a general proofreading of the text before it can be presented to the journal. Some problems have been found (e.g., chorionic, instead of chronic).

ABSTRACT

In this part, what lacks is the type of study and whether the authors use cross-sectional or longitudinal data.

In the end, the authors list a series of differences between groups. However, this aim was not mentioned before.

BACKGROUND

I understand that the topic is important, with a lot of variables on the way, but the introduction needs to be shortened significantly; otherwise, the reader gets boring. 

The paragraphs are in disorder. For example, the p. 1.5 regards the beduin society in Israel, but for some reasons, this contain also the aims of the study, which should not.

I suggest not dividing further the intro with subparagraphs, except for the subparagraph "aims", which is justified.

METHODS

Please report the range scores for each instrument described

I am a bit perplexed about the instrument built for measuring epilepsy severity. Was this tool validated? How did the authors design this? On what evidence? More information on this tool is clearly needed and also references added; the same also for the extent of pharmacotherapy. 

Description of sample size estimation is poor. What is d? Please expand

The statistical methods are incomplete. For example, how did the authors describe the sample, and detected differences across groups? Was it a cross sectional study? If so, whic timepoint was used to perform the path analysis? If not, was it longitudinal? Which timepoints?

RESULTS

In Table 1 I struggle to understand the term "Sx". Please fix it, or add a legend at the botton of the table. Also add the legend for the acronyms used: e.g., SD, n, t.

The statistical analysis has been conducted well. However, the decision regarding the position of the variables in the path analysis should be theoretically justified. Is there a theory underlying how the variables were to be studied (i.e., the relationship between them)? An important point to clarify is why some variables have been chosen as indirect predictors (mediators) and others as indipendent predictors?

Figure 1 needs to be better detailed in the notes at the bottom. For example, are the coefficients standardized? What is the number in brackets? what is the meaning of double headed arrows, and single headed arrows? and so on..

There is no need to repeat the ideal cut offs of the fit indices. Please remove them

Table 2 needs to be explained better in the notes; are these standardized coefficients? For the indirect effects, I do not see the bootstrapped CIs, which are key in this mediation analysis.

Every indirect effect described in the text should be reported alongside the bootstrapped CIs 

Please do not repeat elements of the methods in the results (e.g., sample size estimation).

Graphs should be reported as per international guidelines reporting

DISCUSSION

This section should begin with repeating briefly the aims and the results found in the study.

REFERENCES

Where possible, update the references at not older than 5 years.

moderate proofreading required

Author Response

1.  I suggest a general proofreading of the text before it can be presented to the journal. Some problems have been found (e.g., chorionic, instead of chronic).

All typos have been corrected.

2.  In this part, what lacks is the type of study and whether the authors use cross-sectional or longitudinal data.

We now specify that data are cross-sectional (collected at one point in time).

3.  In the end, the authors list a series of differences between groups. However, this aim was not mentioned before.

Yes, our aim was to identify predictors or burden; however, reported between group differences are also interesting.

4.  I understand that the topic is important, with a lot of variables on the way, but the introduction needs to be shortened significantly; otherwise, the reader gets bored. 

We discuss the literature describing each of the independent variables related to burden associated with epilepsy care.

5.  The paragraphs are in disorder. For example, the p. 1.5 regards the beduin society in Israel, but for some reasons, this contain also the aims of the study, which should not.

We end the introduction by presenting our research question.

7.  I suggest not dividing further the intro with subparagraphs, except for the subparagraph "aims", which is justified.

The introduction is already divided into five logical sections, organized according to independent variables.

7.  Please report the range scores for each instrument described.

Range of scores is reported in Table 1.

8.  I am a bit perplexed about the instrument built for measuring epilepsy severity. Was this tool validated? How did the authors design this? On what evidence? More information on this tool is clearly needed and also references added; the same also for the extent of pharmacotherapy. 

The epilepsy severity and pharmacotherapy measures were developed for this study as objective indices of epilepsy care (e.g., duration of symptoms, extent of treatment).

9.  Description of sample size estimation is poor. What is d? Please expand

Cohen’s d is commonly used to specify statistical power.  Where d = .80, the sample size is sufficient to detect medium to large effect sizes.  We reference the power table for path analyses reported in the text by O’Rourke and Hatcher (2013, p. 115).

10. The statistical methods are incomplete. For example, how did the authors describe the sample, and detected differences across groups? Was it a cross sectional study? If so, which time-point was used to perform the path analysis? If not, was it longitudinal? Which timepoints?

This was a cross sectional study as noted above.  As discussed in the text, we began with a hypothesized model in which all independent variables were assumed to directly predict caregiver burden.

11. In Table 1 I struggle to understand the term "Sx". Please fix it, or add a legend at the bottom of the table. Also add the legend for the acronyms used: e.g., SD, n, t.

A note to Table 1 has been added.

12. The statistical analysis has been conducted well. However, the decision regarding the position of the variables in the path analysis should be theoretically justified. Is there a theory underlying how the variables were to be studied (i.e., the relationship between them)? An important point to clarify is why some variables have been chosen as indirect predictors (mediators) and others as independent predictors?

The data, not theoretical decisions, determined the placement of variables and the direction of variables.

13. Figure 1 needs to be better detailed in the notes at the bottom. For example, are the coefficients standardized? What is the number in brackets? what is the meaning of double headed arrows, and single headed arrows? and so on.

We have added a figure note stating that parameters are expressed as maximum likelihood estimates (standardized solution). Parenthetical numbers indicate significance levels (CR > 1.96, p < 0.05; CR >  2.58, p < .01).

14. There is no need to repeat the ideal cut offs of the fit indices. Please remove them.

Done

15. Table 2 needs to be explained better in the notes; are these standardized coefficients? For the indirect effects, I do not see the bootstrapped CIs, which are key in this mediation analysis.

Table 2 appears directly following the figure to show that each path in the model appears as a direct effect in the table.

16. Every indirect effect described in the text should be reported alongside the bootstrapped CIs 

Our sample size is not sufficient to compute accurate CIs.

17. Please do not repeat elements of the methods in the results (e.g., sample size estimation).

Once the final model is determined (with six independent variables), we confirm the power for this, not the hypothesized, model.

18. Graphs should be reported as per international guidelines reporting

Figures and tables are reported consistent with convention and previously published research in this and other Journals.

19. The discussion should begin with repeating briefly the aims and the results found in the study.

As we state in the first paragraph of the discussion, our goal was to identify both direct and indirect predictors of epilepsy care burden; this was achieved.

20. Where possible, update the references at not older than 5 years.

It is not feasible to restrict references to those published in the past five years.  For instance, well-validated scales used in this study (and supporting psychometric studies) were published earlier.

Reviewer 2 Report

This is an interesting study with a novel population. A few suggestions for improvement include:

1. On page 2, in the second paragraph, "chronic" is misspelled as "chorionic."

2. On page 3, the authors describe how there is much diversity within the Bedouin community, regarding access to resources and rurality of living arrangements.  However, these statistics are not reported for the study's participants, despite the direct impact that they could have on the study's findings.

3. Page 7's discussion opines that Bedouin mothers are reporting greater burden than Jewish mothers due to cultural factors.  However, the findings within the sample indicate that the children of Bedouin mothers had significantly more severe symptomology than those of the Jewish mothers.  Is that not a possible (and probable) factor affecting their perceived burden?

4. Why do the authors assume that the higher severity of symptoms expressed by Beduoin mothers is a measurement error, and that they are under-reporting the less severe cases? Please provide more evidence to substantiate this claim.

Author Response

1.  On page 2, in the second paragraph, "chronic" is misspelled as "chorionic."

Typos have been corrected.

2.  On page 3, the authors describe how there is much diversity within the Bedouin community, regarding access to resources and rurality of living arrangements.  However, these statistics are not reported for the study's participants, despite the direct impact that they could have on the study's findings.

In Table 1, we compare Jewish and Bedouin mothers on each of the study variables appearing in our model (see t tests). Moreover, ethnicity emerged as a significant direct and indirect predictor of caregiver burden.

3.  Page 7's discussion opines that Bedouin mothers are reporting greater burden than Jewish mothers due to cultural factors.  However, the findings within the sample indicate that the children of Bedouin mothers had significantly more severe symptomology than those of the Jewish mothers.  Is that not a possible (and probable) factor affecting their perceived burden?

Note, the emerging path model is multivariate, meaning that ethnicity along with other independent variables are simultaneously examined relative to caregiver burden.  In other words, there are both differences in disease severity as well as psychosocial variables that differ between Jewish and Bedouin mothers.

4.  Why do the authors assume that the higher severity of symptoms expressed by Bedouin mothers is a measurement error, and that they are under-reporting the less severe cases? Please provide more evidence to substantiate this claim.

We do not assume that reporting differences reflect measurement error.  We instead assume cultural differences.  Considerable misconceptions regarding mental illness persist in Bedouin society (e.g., evil spirits, demonic possession).  Unless symptoms are severe, they do not resent to clinicians.  This applies to all psychiatric and movement disorders, not just epilepsy.

Round 2

Reviewer 1 Report

Thanks for this feedback. Although the authors responded to all the queries, I think that some aspects were not adequately addressed. I report the number of the queries with the comment of the reviewer.

1) This observation lies in the fact that the aims should always be consistent with the results presented. Given that the authors presented differences between groups, this aim should also be reported.

5) What the reviewer wanted to point out in this query is that the parapraph relative to the aims should be separated from the rest of the text; now it is merged with the paragraph beduin society in Israel, and this makes nonsense.

7) The reviewer strongly suggests reporting the score ranges in the appropriate section and not in the Tables, which should only contain the results (no elements of instruments description)

12) It is important that the position of the variables (e.g., mediators, IV and DV) are determined both empirically and theoretically. If that was not done, this is an important limitation of the study and needs to be acknowledged in the appropriate section

13) Figure 1 is still confusing. Please, report the standard errors in brackets and asterisks to indicate p values.

16) This reviewer does not understand why the sample size is insufficient to report bootstrapped Cis. With n=130, there should not be any problems at all. These confidence intervals are very important to report in every mediation analysis, because the distribution of the indirect effects does not follow gaussian distribution, and therefore the significance of the results is not reliable at all. If the authors are not able to report BCI the results cannot be presented in a publication.

18) Please remove the colors in the figure; insert the standard errors in brackets, and use the asterisks and notes at the bottom of the table to indicate significance of the parameters. I suggest removing this part: (CR >½1.96½, p < 0.05; CR > ½2.58½, p < .01), which is confusing to the reader.

19) As it was suggested, please begin with repeating the aims and the results briefly. This was not achieved, since the authors began with…Caring for a child with epilepsy is often difficult, requiring vigilance and associated 255 stress. This leads many to feel burdened. Demands of care are even more difficult for eth- 256 nic minority families with fewer resources and structural barriers.

In brief, there are two important limitations of this study, which if not addressed, may hinder publication: theoretical justification for the choice of the variables in the mediation model, and bootstrapped confidence intervals.

minor english editing required

Author Response

1.  This observation lies in the fact that the aims should always be consistent with the results presented. Given that the authors presented differences between groups, this aim should also be reported.

As ‘ethnicity’ is a primary variable in our path model, this aim is achieved.  Ethnicity is a direct predictor of symptoms of depression, anxiety and caregiver burden as we report in the text. Additional t tests are performed to show differences across independent variables.  That is, univariate analyses support multivariate findings.

2.  What the reviewer wanted to point out in this query is that the paragraph relative to the aims should be separated from the rest of the text; now it is merged with the paragraph Bedouin society in Israel, and this makes nonsense.

It is standard procedure to end the introduction with a statement of research question/s.  As this is a single sentence, it cannot stand alone as a sentence. 

3.  The reviewer strongly suggests reporting the score ranges in the appropriate section and not in the Tables, which should only contain the results (no elements of instruments description)

Although redundant, we now report range of scores in the text as well as Table 1.

4.  It is important that the position of the variables (e.g., mediators, IV and DV) are determined both empirically and theoretically. If that was not done, this is an important limitation of the study and needs to be acknowledged in the appropriate section

The reviewer is conflating path analyses with moderation/mediation analyses; these are related but distinct statistical procedures (i.e., both extensions of multiple regression). As we describe in the text, path analyses for this study were performed in keeping with convention: 1) We tested a hypothesized model (not shown) in which all independent variables were assumed to be direct predictors of caregiver burden; 2) all nonsignificant paths were deleted, and 3) new statistically significant paths were added when supported by theory.

We recommend that the reviewer consult the text by O’Rourke and Hatcher (2013) in which chapter 4 deals specifically with path analysis (see link below).  The procedures followed in this paper fully conform to the steps described by O’Rourke and Hatcher (2013).

5.  Figure 1 is still confusing. Please, report the standard errors in brackets and asterisks to indicate p values.

Again, as described by O’Rourke and Hatcher (2013), it is convention to report only standardized (or unstandardized) path coefficients, confidence ratios (CR) and associated significance levels.  Confidence intervals with smaller samples are not reported in path analysis.

Note that the presentation of findings is consistent with other papers published in top quartile journals (https://formative.jmir.org/2023/1/e44059; http://doi:10.1177/0886260520908022)

6.  This reviewer does not understand why the sample size is insufficient to report bootstrapped Cis. With n=130, there should not be any problems at all. These confidence intervals are very important to report in every mediation analysis, because the distribution of the indirect effects does not follow gaussian distribution, and therefore the significance of the results is not reliable at all. If the authors are not able to report BCI the results cannot be presented in a publication.

With small sample sizes (under 200) confidence intervals are very wide and not interpretable. This change will not be made. As noted above, these are not mediation analyses (instead, direct and indirect path analyses).

7.  Please remove the colors in the figure; insert the standard errors in brackets, and use the asterisks and notes at the bottom of the table to indicate significance of the parameters. I suggest removing this part: (CR >½1.96½, p < 0.05; CR > ½2.58½, p < .01), which is confusing to the reader.

Use of colour in path and structural equation models is commonly used, including papers recently published in this journal (see https://dx.doi.org/10.3390/healthcare11131965).  This fosters the interpretability of findings and makes it more readable (i.e., will not be removed). It is convention to report confidence ratios (not confidence intervals) associated with each beta value (see O’Rourke & Hatcher, 2013).  We have, however, removed absolute value signs as all beta values are positive.

8.  As it was suggested, please begin with repeating the aims and the results briefly. This was not achieved, since the authors began with…Caring for a child with epilepsy is often difficult, requiring vigilance and associated stress. This leads many to feel burdened. Demands of care are even more difficult for ethnic minority families with fewer resources and structural barriers.

We see no need to make stylistic changes the text.

9.  In brief, there are two important limitations of this study, which if not addressed, may hinder publication: theoretical justification for the choice of the variables in the mediation model, and bootstrapped confidence intervals.

Variables used in this model are those that have previously been identified as predictors of burden in this and other populations; this is described in the literature review.  And importantly, we computed and report path analyses, not mediation/moderation analyses.

O’Rourke, N., & Hatcher, L. (2013).  A Step-by-Step Approach to Using SAS for Factor Analysis and Structural Equation Modeling (2nd Ed.).  Cary, NC: SAS Institute.

Author Response

No further edits/revisions requested